# Predictive Factors Correlated with Successful Early Endoscopic Removal of Pancreaticolithiasis in Chronic Pancreatitis after Extracorporeal Shock Wave Lithotripsy

**DOI:** 10.3390/diagnostics14020172

**Published:** 2024-01-12

**Authors:** Thanawin Wong, Tanawat Pattarapuntakul, Nisa Netinatsunton, Bancha Ovartlarnporn, Jaksin Sottisuporn, Thanapon Yaowmaneerat, Siriboon Attasaranya, Kamonwon Cattapan, Pimsiri Sripongpun

**Affiliations:** 1Gastroenterology and Hepatology Unit, Division of Internal Medicine, Faculty of Medicine, Prince of Songkla University, Hat Yai 90110, Songkhla, Thailand; wongthanawin68@gmail.com (T.W.); spimsiri@medicine.psu.ac.th (P.S.); 2NKC Institute of Gastroenterology and Hepatology, Songklanagarind Hospital, Faculty of Medicine, Prince of Songkla University, Hat Yai 90110, Songkhla, Thailand; nisasan@yahoo.com (N.N.); obancha@live.com (B.O.); pondjaksin@hotmail.com (J.S.); thanapon.y@psu.ac.th (T.Y.); sattasar@gmail.com (S.A.); 3Department of Radiology, Faculty of Medicine, Prince of Songkla University, Hat Yai 90110, Songkhla, Thailand; kamonwon.c@psu.ac.th

**Keywords:** Pancreaticolithiasis, pancreatic calculi, chronic pancreatitis, extracorporeal shockwave lithotripsy, endoscopic retrograde cholangiopancreatography, endoscopic clearance, predictive factors, pancreatic stone density

## Abstract

Background: The treatment of chronic pancreatitis (CP) and symptomatic pancreatic duct (PD) calculi often involves techniques like endoscopic retrograde cholangiopancreatography (ERCP), extracorporeal shock wave lithotripsy (ESWL), or a combination of both. However, identifying predictive factors for the successful removal of these calculi remains variable. This study aimed to determine the factors predicting successful ESWL and endoscopic removal in CP and PD calculi patients. Methods: We examined data from CP patients who underwent complete PD calculi removal via ESWL combined with ERCP between July 2012 and 2022, and assessed baseline characteristics, imaging findings, and treatment details. Patients were categorized into early- and late-endoscopic complete removal groups (EER and LER groups, respectively). Results: Of the 27 patients analyzed, 74.1% were male with an average age of 44 ± 9.6 years. EER was achieved in 74% of the patients. Patients in the EER group exhibited smaller PD calculi diameter (8.5 vs. 19 mm, *p* = 0.012) and lower calculus density (964.6 vs. 1313.3 HU, *p* = 0.041) compared to the LER group. Notably, PD stricture and the rate of PD stent insertion were not different between the groups. A calculus density threshold of 1300 HU on non-contrast CT demonstrated 71% sensitivity and 80% specificity in predicting EER. Conclusions: Smaller and low-density PD calculi may serve as predictors for successful EER, potentially aiding in the management of CP patients with PD calculi.

## 1. Introduction

Chronic pancreatitis (CP) is a progressive, inflammatory disease, characterized by pancreatic atrophy, fibrosis, ductal stricture, calcification, stone formation, exocrine insufficiency, diabetes mellitus, and abdominal pain [1]. CP significantly increases the risk of developing pancreatic ductal adenocarcinoma [2]. While alcohol remains the predominant cause in many developed nations, non-alcoholic idiopathic CP prevails in other regions, particularly among Asian populations. The highest prevalence has been reported in India [3,4]. The intricate interplay between genetic predispositions and environmental influences plays a substantial role in the complex pathogenesis of CP [5]. Abdominal pain serves as the most prevalent symptom among CP patients, varying between 60% and 94% in different studies [6,7,8]. Individuals experiencing an early onset of the disease and those linked to an alcohol-related cause are predisposed to a higher likelihood of experiencing pain associated with CP. Despite this correlation, the precise mechanisms triggering such pain sensations remain elusive [9]. The manifestations and patterns of pain in CP exhibit variability based on the temporal progression and severity of the condition. The quality of life of affected individuals is significantly impacted by the frequency and intensity of pancreatic pain [10].

Pancreaticolithiases, also known as pancreatic calculi (PC), represent a consequence of CP and can manifest within the primary duct, side branches, or pancreatic parenchyma, regardless of the underlying cause of CP. These calculi are a result of stagnant secretions and the calcification of protein plugs. Typically, PC comprises an inner core, or nidus, encompassed by successive layers of calcium carbonate. They are categorized based on their characteristics, quantity, and location within the pancreas. These classifications encompass the calculi’s radiopacity (appearing radiopaque, radiolucent, or a mixture); quantity (single or multiple); positioning within the main pancreatic duct (MPD), side branches, or the pancreatic parenchyma; and their location within the head, body, or tail of the pancreas [11]. The majority of PC demonstrate radiopaque properties. A potential explanation for the pain experienced in chronic pancreatitis involves the obstruction of the pancreatic duct (PD) resulting from the presence of stones, strictures, or a combination of both, leading to elevated pressure within the duct. It is worth noting that PD calculi are prevalent, occurring in approximately 90% of CP patients [12,13,14]. The primary treatment protocol for alleviating chronic abdominal pain involves the complete removal of PD calculi. To address this, various approaches are employed, including medical therapy to manage the pain, along with both endoscopic therapy (ET) and surgical interventions. These procedures are intended to clear the calculi and reduce the pressure within the duct, aiming to alleviate the associated symptoms [15]. 

Regarding PC measuring less than 5 mm, the preferred treatment method includes employing endoscopic retrograde cholangiopancreatography (ERCP), followed by the sphincterotomy on either the major or minor papilla, or both. This procedure typically involves the temporary placement of stents or the extraction of the calculi from the pancreas using a balloon or basket. However, calculi larger than 5 mm in size are often impacted and are difficult to extract using the standard techniques. These calculi must be fragmented to facilitate extraction [15]. Extracorporeal shock wave lithotripsy (ESWL) is based on the principles of either electrohydraulic or electromagnetic shock wave energy. These shock waves are precisely directed towards the specified calculi, employing an acoustic lens or cylindrical reflector. Upon impact, the shock waves create a compressive force along the surface of the calculus, ultimately resulting in the fragmentation of the calculi into powdered form [16]. This innovative technique aims to disintegrate the targeted calculi into smaller particles for easier extraction or elimination. The guidelines outlined by the European Society of Gastrointestinal Endoscopy in 2018 advocate the utilization of ESWL as the primary approach for clearing radiopaque MPD calculi that exceed 5 mm in size and are situated within the pancreatic head and body [17]. ESWL can be conducted either independently or in combination with ERCP. This method’s effectiveness in fragmenting calculi reaches up to 90%, and it leads to complete clearance of the calculi in 80% of cases. This procedure is deemed highly efficient for disintegrating and ultimately eliminating calculi, particularly those of specified size and location within the pancreas [11,18]. This treatment approach contributes significantly to enhancing the patient’s overall quality of life by alleviating pain, minimizing the need for narcotics use, and ultimately reducing the necessity for surgical interventions in the pancreas [19].

Reportedly, there are various factors, such as duct diameter, size of the calculus, MPD stricture, receipt of pre-ESWL ERCP, and PD calculi density, that result in the outcome of ESWL and endoscopic removal of PD calculi [20,21]. Cases in which the PD is dilated more than 8 mm in diameter and in which PD calculi are greater than 12 mm in size are associated with the unsuccessful endoscopic removal of calculi following the ESWL session [22]. Alternative ET methods are employed to eliminate large PD calculi, such as pancreatoscopy-guided electrohydraulic lithotripsy (EHL) and laser lithotripsy. However, pancreatoscopy-guided EHL displays inconsistent rates of fragmentation and success [23,24]. Moreover, it is associated with post-procedural pancreatitis in approximately 28% of cases [25]. Surgical interventions, including resection and drainage, are reserved for patients with extensive calculi, multiple strictures, suspected pancreatic carcinoma, or those who have not responded well to ET methods. Several factors are associated with unfavorable ESWL outcomes in the complete powderization and clearance of calculi within the duct. These include the presence of multiple and larger calculi, as well as calculi associated with strictures in the MPD. However, there are scarce data regarding factors predicting successful treatment with a combination of ESWL and ERCP; moreover, there are limited reports from Thailand, which has a different population and may, therefore, attribute CP to different etiologies. This research aims to identify the predictive factors that determine the technical success of the ESWL in combination with endoscopic removal in treating patients with CP and PC.

## 2. Materials and Methods

### 2.1. Study Design 

The study protocol was approved by the human research ethics committee (HREC) of the Faculty of Medicine, Prince of Songkla University, Thailand; REC.65-390-14-3.

We conducted a retrospective cohort study of all patients with obstructive and painful CP who were scheduled for removal of PD calculi using ESWL and ERCP at our center, the Nanthana-Kriangkrai Chotiwattanaphan (NKC) Institute of Gastroenterology and Hepatology between July 2012 and July 2022. The NKC Institute is located in Songklanagarind Hospital, which is the only university hospital in Southern Thailand. The medical records and baseline characteristics, imaging findings, and pre-ESWL ET of each patient were reviewed and evaluated by a single investigator. Of all 35 patients in the study period, eight patients were excluded from the analysis, as five patients had failed PD cannulation, and three patients had no available baseline cross-sectional imaging before the primary procedural treatment to be reviewed. In addition, patients with past medical histories of pancreatic surgery or a diagnosis of pancreatic cancer were not included in the statistical analysis. Finally, the data of 27 patients were statistically analyzed (Figure 1).

The characteristics, location, and duct diameter of the calculi were determined using non-contrast computed tomography (NCCT) and reviewed by a radiologist experienced in abdominal radiology. The slice thickness was set at 5 mm. Images that revealed the calculi at their largest dimensions were considered for analysis. Attenuation values in Hounsfield units (HU) were measured by drawing a region of interest over the calculus on axial NCCT images by a radiologist blinded to the patients’ other clinical data. 

Pre-ESWL ERCP was defined as a pancreatic sphincterotomy, PD stenting, PD stricture dilation, or retrieval of calculi prior to the first ESWL session. The decision of whether patients underwent pre-ESWL ERCP or not was made by the primary endoscopist. 

PD stent placement was defined as any placement of a PD stent either before or after ESWL ERCP procedures.

Early endoscopic removal (EER) was defined as no more than three ERCP attempts to complete the removal of the MPD calculi. Late endoscopic removal (LER) was defined as four or more attempts at ERCP to complete remove the MPD calculi.

Complete removal of the MPD calculus was defined as no filling defect in the MPD on pancreatogram during ERCP.

### 2.2. Extracorporeal Shock Wave Lithotripsy

All patients underwent a series of regular and consecutive sessions of ESWL until the largest calculus was fragmented to a size of 5 mm or smaller. ESWL procedures were conducted using a Storz Modulith SLX-F2 lithotripter (Storz Medical, Kreuzlingen, Switzerland) while the patient was in a prone position under general anesthesia. The ESWL was performed by a skilled gastroenterologist and technician. To aid in the precise localization of the pancreatic duct (PD) calculus, fluoroscopy guidance was employed along with spinal landmarks and PD stents (see Figure 1). The ESWL treatment spanned multiple days, encompassing repeated sessions lasting between 60 and 90 min each, with a shock frequency of 3000–5000 per session, and an intensity scale rating of 5–6 (15,000–16,000 kV) on a scale from 1 to 6. The number of ESWL sessions conducted for each patient was meticulously documented.

### 2.3. Post-ESWL ERCP

Following the successful fragmentation of the calculus to a size of 5 mm or smaller through ESWL, the patients underwent ERCP utilizing a therapeutic side-viewing duodenoscope (Olympus, Exera TJF 160/180/190, Tokyo, Japan) to ensure the comprehensive elimination of the calculus. Pancreatic sphincterotomy was performed in all patients. The subsequent steps, including the use of balloon or basket extraction, placement of PD stents, dilation of strictures, and the determination of stent size and length, were determined based on the discretion of the endoscopist (Figure 2).

### 2.4. Statistical Analysis

The patients were categorized into either EER or LER cohorts based on the definition described earlier. Comparisons of continuous variables between the two groups were performed using the Wilcoxon test for non-normally distributed data and the Student’s *t*-test for normally distributed data. Categorical data were compared using the chi-square test or Fisher’s exact test. Statistical significance was set at a *p* < 0.05. The logistic regression analysis was used to define factors predictive for EER. Independent risk factors were expressed as odds ratios (ORs) with 95% confidence intervals (CIs). The optimal density cutoff value of the calculi that differentiated EER from LER was determined using the receiver operating characteristic (ROC) curve analysis, and the area under the curve (AUC) was calculated. All statistical analyses were performed using R software version 4.2.1 (R Foundation for Statistical Computing, Vienna, Austria).

## 3. Results

Table 1 shows the baseline characteristics and interventions of the patients. Of the 27 patients, 74.1% were male (*n* = 20), with a mean age of 44 ± 9.6 years. Sixteen (59.3%) of the patients with CP had alcohol-induced pancreatitis. Pre-ESWL ERCP was performed in 55.6% (*n* = 15) of the patients. PD strictures were found in 66.7% (*n* = 18) of the patients. The total number of ESWL sessions had a median of 2 (interquartile range 2–4). The number of shocks ranged from 3000 to 5000 shocks with a median of 4501. Multiple calculi were observed in 77.8% of the patients.

There were no statistically significant differences between the EER and LER groups in terms of age, body mass index, and etiology of chronic pancreatitis (*p* = 0.963, *p* = 0.443, and *p* = 0.902, respectively). In addition, there were no statistically significant differences in the number of interventions performed between the two cohorts (*p* = 0.408, *p* = 1, and *p* = 0.269, for pre-ESWL ERCP, PD stricture, and PD stent placement, respectively). PD stents were placed in 23/27 (85.2%) patients. All PD stents were placed during the first ERCP, whether in patients who underwent pre-ESWL ERCP (15/15) or in those who did not undergo pre-ESWL ERCP, PD stents were placed during the first post-ESWL ERCP in the latter group. Furthermore, there was no statistically significant difference in the number of shocks per session between the two cohorts (*p* = 0.749). Foreseeably, the total number of post-ESWL ERCP procedures in the EER group was significantly lower than in the LER group (2 vs. 4, *p* < 0.001)

Table 2 summarizes the imaging findings and characteristics of the calculi. The median maximum diameter of the calculi in the LER cohort was statistically significantly larger than that in the EER cohort (19 vs. 8.5 mm, *p* = 0.012). The mean stone density in the LER cohort was statistically significantly higher than that in the EER cohort (964.6 vs. 1313.3 HU, *p* = 0.041). When using the threshold cutoff at 1300 HU, the calculus density on non-contrast CT revealed a sensitivity of 71% and specificity of 80% in predicting EER. The ROC curve analysis for PD calculus density revealed an AUC of 0.757 (Figure 3. There were no statistically significant differences between the two cohorts in the other findings, including location of the calculus, number of calculi, MPD size, and splenic vein thrombosis.

Based on the univariate analysis, EER was more frequent when the calculus density < 1300 HU (OR = 10 [95% CI = 1.39–71.86]) (Table 3). When the calculus density threshold was set at 1300 HU, EER was achieved in 80% (*n* = 16) of the patients with lower-density calculi. The maximum PC size of <10 mm was also significantly associated with the EER (OR = 14 [95% CI = 1.37–142.89]). PD stricture, stent placement, an etiology of alcohol, as well as pre-ESWL ERCP were not significantly associated with EER.

The complications related to endoscopic therapy were noted in 3 out of 27 (11.1%) patients, including mild acute pancreatitis (*n* = 1), abdominal pain (*n* = 1), and minor bleeding (*n* = 1). Complications related to ESWL were noted in 1 out of 27 (3.7%) patients, presenting as mild acute pancreatitis. All patients with complications recovered following conservative therapy and no severe complications were noted.

## 4. Discussion

CP is an inflammatory disease that gradually damages the pancreatic tissue and its duct structures. Clearing intraductal stones entirely can offer both immediate and lasting relief from pain. In the case of a large and impact MPD stone, fragmenting these stones within the PD might be necessary to ease their endoscopic removal, considering the challenges involved in such removal attempts. ESWL has an efficiency rate of 90% in stone fragmentation and can result in achieving complete clearance of the MPD up to 80% [12,13,14]. However, the achievement of complete stone clearance varied based on several factors and remained a subject of ongoing debate.

Our study was conducted at a tertiary university hospital and involved individuals diagnosed with CP and pancreatic duct PD calculi who underwent both ESWL and ERCP, resulting in the complete elimination of the calculi.

Approximately 60% of study patients had chronic alcoholic pancreatitis, identified as the most frequently observed cause of CP. There were no statistically significant differences in the other baseline characteristics between the two cohorts. Remarkably, in our study, only the size of the calculus, not the diameter of the duct, correlated with an increased number of ESWL sessions and endoscopic removal efforts. Calculi measuring less than 10 mm in size necessitated no more than three attempts of ERCP for removal. The diameter of MPD and the presence of strictures did not serve as predictive factors for EER. These findings align with a study conducted in Japan, emphasizing similarities in the clinical outcomes related to the treatment of pancreatic calculi [26].

The current study demonstrated no association between pre-ESWL ERCP and the placement of PD stents with the effective endoscopic removal of calculi. This lack of correlation might be due to the relatively small size of our patient cohort; however, it mirrored the outcomes observed in prior published studies, which indicated that performing ERCP before ESWL and the placement of PD stents do not significantly contribute to the successful clearance of PD calculi [22].

Only the findings of a study by Ohyama et al. revealed the efficacy of measuring calculus density using NCCT in the prediction of therapeutic ESWL outcomes. Ohyama et al. found that a threshold of 820.5 HU of PD calculi indicated complete removal of calculi in almost 80% of cases [26]. Our current study contributed additional insights into measuring calculus density using NCCT. A calculus density below 1300 HU predicted the achievement of EER, demonstrating a sensitivity of 71.8% and a specificity of 80%. Utilizing NCCT, an affordable and accessible tool, aids endoscopists in anticipating the outcomes of endoscopic removal. Moreover, carrying out ERCP without a baseline NCCT raises the risk of post-ERCP pancreatitis and reduces the rate of ductal clearance [27]. In settings with limited resources, employing baseline NCCT to accurately foresee the successful endoscopic removal of calculi using ESWL, without initial pre-ESWL ERCP attempts, proves valuable in determining treatment outcomes.

Complications of ESWL and ET were reportedly approximately 4–7%, with post-ESWL pancreatitis being the most prevalent among these issues [19,28,29,30,31].

To date, there have been limited publications addressing the predictive factors influencing endoscopic removal utilizing ESWL, ERCP, or a combination of both methods. A complete clearance rate of calculi was only established in a systematic review and meta-analysis [32]. Furthermore, the predictive factors were only observed in retrospective studies. While the insufficiency of data from randomized control trials remains evident. This is the first study that reported on the predictive factors for successful EER after ESWL in treating patients with CP and PC. The study encountered several limitations. As the study followed a single-center, retrospective design, there might be potential biases in treatment selection, as the choice of pre-ESWL ERCP was contingent upon the discretion of the endoscopists. Additionally, the frequency of post-ESWL ERCP procedures and the utilization of mechanical lithotripters were reliant on the endoscopist’s judgment. There might have been variations in the ESWL equipment used in our study compared to other research, albeit our study maintained consistency by using the same equipment from 2012 to 2022. Lastly, we acknowledge that the study sample size was relatively small; however, we included all the patients in a 10-year period at a tertiary care center, representing real-world patients in Southern Thailand, and we believed that the results provided some insightful information for such patients.

## 5. Conclusions

The current study revealed a notable correlation between the size and density of calculi and their impact on the success of endoscopic removal. Pre-ESWL ERCP, PD stricture, and PD stent placement did not exert influence on the clearance of stones in the MPD. Additionally, the predictive nature of calculi smaller than 10 mm and with a density lower than 1300 HU, as identified through baseline NCCT, were indicative of EER following ESWL. These insights provide substantial advantages in managing patients with CP in Thailand and offer essential guidance for tailoring treatment strategies for selected patient groups.

## Data Availability

The datasets used and/or analyzed during the current study are available from the corresponding author on reasonable request.

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
