# Peer review of "Predictive Factors Correlated with Successful Early Endoscopic Removal of Pancreaticolithiasis in Chronic Pancreatitis after Extracorporeal Shock Wave Lithotripsy"

_diagnostics, 2024, doi:10.3390/diagnostics14020172_

Round 1
Reviewer 1 Report
Comments and Suggestions for Authors
I want to congratulate you for your courage. It is a new technique that needs to be developed but that requires a good selection of cases.
Author Response
Thank you for your comment and suggestion
Reviewer 2 Report
Comments and Suggestions for Authors
Dear authors,
congratulations for this very good original article. I advise only minor grammatical corrections.

Author Response
Thank you for your comment and suggestion. We have corrected the grammatical errors accordingly.
Reviewer 3 Report
Comments and Suggestions for Authors
Thank you for giving me an opportunity to review this article. The authors have reported the “Predictive factors Correlated with Successful Early Endoscopic Removal of Pancreaticolithiasis in Chronic Pancreatitis after Ex-tracorporeal Shock Wave Lithotripsy”. This article is well-written, however, there are few commend.
Major command:
1. Sample size was too small and two groups were not equal.
2. Why did author definite the early endoscopic removal (EER) as no more than three ERCP attempts to complete removal of the MPD calculus? Was the early endoscopic removal (EER) defined as no more than two ERCP better than three ERCP?
3. Why didn’t the author record the procedure time in each ERCP or group? Could the author record and put in Table1?
4. Could the author give the complication about these ERCP procedures including bleeding, pancreatitis, and abdominal pain?
5. Could the author explain the type of ERCP procedure for Naïve papilla (EPBD or EPT)?
6. Did the author put the P-duct in 1st time ERCP or when did the author put the P-duct stent?

Minor editing of English language required
Author Response
1. Sample size was too small and two groups were not equal.
- Thank you, we acknowledge the small sample size as you mentioned. However, we’ve put our best effort to include all the eligible cases treated at our hospital, which is the only tertiary university hospital in Southern Thailand for the entire 10-year period. And we have stated the small sample size as a limitation in discussion part already (as highlighted on page 9)
2. Why did author definite the early endoscopic removal (EER) as no more than three ERCP attempts to complete removal of the MPD calculus? Was the early endoscopic removal (EER) defined as no more than two ERCP better than three ERCP?
- Thank you for pointing this out, there is no consensus definition for early or late endoscopic removal. In this study, we arbitrarily chose the number of ERCP sessions of <3 to define EER according to the study by Tadenuma et al. In that study, of a total 117 patients, the median number of endoscopic sessions for complete removal was 3.0 +2.0 sessions, therefore, we use <3 sessions to determine EER and LER.
3. Why didn’t the author record the procedure time in each ERCP or group? Could the author record and put in Table1?
- The procedure time recorded in the hospital information system encompasses the time for the entire ERCP procedure. It not only includes endoscopic retrograde pancreatography with P-duct stone removal but also incorporates the duration of distal CBD stricture treatment in some cases. As the retrospective nature of the study, the exact time for the procedure spent on pancreatic calculi could not be retrieved. Therefore, we believe that the procedure time recorded in the systemmay not accurately represent the specific process of P-duct stone removal and was not included in the analysis.
4. Could the author give the complication about these ERCP procedures including bleeding, pancreatitis, and abdominal pain?
- Thank you, we have added the results regarding complication as you suggested. The complications related to endoscopic therapy were noted in 3 out of 27 (11.1%) patients, including mild acute pancreatitis in 1 case, abdominal pain in 1 case, and minor bleeding in 1 case. Complications related to ESWL were noted in 1 out of 27 (3.7%) patients, presenting as mild acute pancreatitis. This was added to the results part on page 8 as highlighted.
5. Could the author explain the type of ERCP procedure for Naïve papilla (EPBD or EPT)?
- In our study, all cases underwent endoscopic pancreatic sphincterotomy, and 5 out of 27 patients underwent endoscopic sphincterotomy (EPT) for CBD cannulation. No cases underwent endoscopic papillary balloon dilatation (EPBD) because most CBD pathologies in chronic pancreatitis involved distal CBD strictures, unlike cases with large CBD stones. The statement regarding endoscopic pancreatic sphincterotomy was added in the materials and methods part as highlighted on page 4.
6. Did the author put the P-duct in 1st time ERCP or when did the author put the P-duct stent?
- In our study, P-ducts stents were placed in 23/27 (85.2%) patients. All P-duct stents were placed during the first ERCP, whether in patients who underwent pre-ESWL ERCP (15/15) or in those who did not undergo pre-ESWL ERCP. P-duct stents were placed during the first post-ESWL ERCP in the latter group. The statement regarding P-duct stents was added as highlighted in the results part on page 8.